# Prompt Resumption of Screening Programme Reduced the Impact of COVID-19 on New Breast Cancer Diagnoses in Northern Italy

**DOI:** 10.3390/cancers14123029

**Published:** 2022-06-20

**Authors:** Lucia Mangone, Pamela Mancuso, Maria Barbara Braghiroli, Isabella Bisceglia, Cinzia Campari, Stefania Caroli, Massimiliano Marino, Adele Caldarella, Paolo Giorgi Rossi, Carmine Pinto

**Affiliations:** 1Epidemiology Unit, Azienda Unità Sanitaria Locale-IRCCS di Reggio Emilia, 42122 Reggio Emilia, Italy; pamela.mancuso@ausl.re.it (P.M.); mariabarbara.braghiroli@ausl.re.it (M.B.B.); isabella.bisceglia@ausl.re.it (I.B.); paolo.giorgirossi@ausl.re.it (P.G.R.); 2Cancer Screening Unit, Azienda Unità Sanitaria Locale–IRCCS di Reggio Emilia, 42122 Reggio Emilia, Italy; cinzia.campari@ausl.re.it (C.C.); stefania.caroli@ausl.re.it (S.C.); 3Department of Clinical Governance, Azienda Unità Sanitaria Locale–RCCS di Reggio Emilia, 42122 Reggio Emilia, Italy; massimiliano.marino@ausl.re.it; 4Tuscany Cancer Registry, Clinical and Descriptive Epidemiology Unit, Institute for Cancer Research, Prevention and Clinical Network (ISPRO), 50139 Florence, Italy; a.caldarella@ispro.toscana.it; 5Medical Oncology Unit, Comprehensive Cancer Centre, Azienda Unità Sanitaria Locale-IRCCS di Reggio Emilia, 42122 Reggio Emilia, Italy; carmine.pinto@ausl.re.it

**Keywords:** COVID-19, breast cancer, incidence, screening, treatment, stage, population-based

## Abstract

**Simple Summary:**

The aim of this study was to compare 2020 tumours with 2019 tumours by age, stage and treatment in four different periods. In 2020 there was no decrease of invasive tumours nor in situ (513 vs. 493 and 76 vs. 73, respectively), while there was a significant decrease in surgery and an increase in neoadjuvant chemotherapy (*p* = 0.016). During the Italian lockdown period (March–May), we observed a decrease in all ages and a significant one among people aged 75+ [IRR 0.45 (95% CI 0.25–0.79)], but in the *last period* there was a significant increase among people of the screening age range of 45–74 [IRR 1.48 (95% CI 1.11–1.98)]. Screening activities were suspended from March to May, but over the summer and the autumn the backlog was eliminated. This suggests that a prompt resumption of programmed screening may have limited the impact of the pandemic on the delay of breast cancer diagnoses.

**Abstract:**

The aim of this study is to evaluate the real impact of COVID-19 during the entire 2020 period, compared with 2019. The data comes from a Cancer Registry in Northern Italy and we compared clinical and treatment characteristics of breast cancer by age, stage, treatment, and status screening. In 2020 there was no decrease in invasive tumours nor in in situ (513 vs. 493 and 76 vs. 73, respectively), while there was a significant decrease in surgery and increase in neoadjuvant chemotherapy (*p* = 0.016). In the screening range (aged 45–74), no change in stage and grading was observed. In the four periods examined there was an increase in new diagnoses during *pre-lockdown*, a decrease in tumours especially at age 75+ [IRR 0.45; 95%CI 0.25–0.79] during *lockdown*, a recovery of new diagnoses in women 45+ in the *low incidence* period while in the *last period* there was a significant increase only for ages 45–74 [IRR 1.48; 95% CI 1.11–1.98]. Screening activities were suspended from March to May, but over the summer and autumn the backlog was addressed. This suggests that a prompt resumption of programmed screening may have limited the impact of the pandemic on the delay of breast cancer diagnoses.

## 1. Introduction

Since the beginning of 2020, SARS-CoV-2 infection has spread dramatically around the world [1]. A national lockdown was introduced in several countries, including Italy, with the aim of containing the advance of the pandemic [2]. These restrictions have changed the daily routine of professionals and, especially during the Italian lockdown (March–May), have been associated with a reduction or suspension of non-COVID related healthcare services [3]. Healthcare systems have responded to the pandemic by trying to reorganize and adapt the allocation of healthcare resources and reorganize staff and infrastructure to minimize the risk of patient exposure, especially in the oncology field [4]. On the other hand, the same oncology departments and healthcare professionals have been called upon to deal with the emergencies of COVID-19 patients. The Italian Association of Medical Oncology has also provided guidelines for the prioritization of breast cancer treatments during the pandemic [5,6]. Various national and international associations and research groups have published the strategies implemented to ensure the continuity of treatment during the COVID-19 pandemic [7,8,9].

All of this healthcare reorganization has had a strong impact on new cancer diagnoses of tumours in general [10,11] and of cancers screened in particular [12,13]: in Italy, Ferrara and colleagues showed a 38.2% decrease in new breast cancer diagnoses [14], which was subsequently confirmed [15].

A major determinant of breast cancer diagnosis reduction was the interruption of mammography screening: in Italy, the National Screening Monitoring Centre reported a 26.6% decrease in invitations and a 37.6% decrease in mammograms performed by the screening programmes in 2020 compared to 2019 [16,17].

The suspension of screening resulted in a decline, especially in the early stages (in situ and stage I) [18,19], but there appears to have been no increase in the incidence of advanced stages and no impact on non-screen detected tumours [20].

The aim of this work is to describe, using population data, the impact of the pandemic on new breast cancer diagnoses for the whole of 2020 compared to 2019, by stage and treatment, in a province heavily affected by the pandemic and with a high participation in mammography screening.

## 2. Materials and Methods

### Case Selection

The study was conducted on 1159 cases of female breast cancer registered in the Reggio Emilia Cancer Registry (RE-CR). The main sources of information are anatomic pathology reports, hospital discharge records, and mortality data integrated with laboratory tests, diagnostic reports, and information from general practitioners. The RE-CR covers a population of about 530,000 inhabitants and is characterized by good data quality (98.8% of microscopic confirmations and 0.1% of Death Certificate Only) [21,22]. The activity of RE-CR was approved by the provincial Ethics Committee of Reggio Emilia, Protocol n. 2014/0019740, on 4 August 2014.

In Emilia Romagna, mammography screening offers an annual mammogram for asymptomatic women aged 45–49 and a biennial mammogram for women aged 50–74. In the province of Reggio Emilia, the participation rate for mammography screening is 76.2%, which is much higher than the national average [23].

In 2020, the screening was interrupted from the end of February to the end of May: for this reason, we divided the period into four intervals: *pre-lockdown* (January–February), *lockdown* (March–May), *low incidence* (June–September), and *second wave* (October–December). In the present study we analyzed the characteristics of women with breast cancer diagnosed from January 2019 to December 2020. 

In particular, we analyzed the age at diagnosis (<45 years, 45–74 and 75+), morphology (ductal, lobular, others), pT (T1 a, b, c, T2, T3, T4), N (0, N+), M (0, 1), stage (in situ, I, II, III, IV), grading (1, 2, 3), and screening status (screen detected, interval cancers, i.e., women with a non-screen detected cancer diagnosed after a negative screening test, non-attenders, not eligible) for women in the screening target age (45–74). We have also added information on the treatment (surgery, adjuvant and neo-adjuvant therapy), with or without hormonal treatment. Descriptive analyses of patient characteristics and characteristics of the cancer were performed by year of diagnosis and age. The association between clinical and demographic variables and year of diagnosis was evaluated through Pearson’s chi-squared *p*-values. Finally, an IRR (Incidence Rate Ratio) was calculated for all four periods of 2020 by age group, i.e., the pre-screening age (<45), the screening target age (45–74) and the post-screening age (75+), compared with 2019, using as a denominator the age specific resident population on 1 January 2020 and 1 January 2019 for the pandemic and pre-pandemic periods, respectively. IRR and relative 95% confidence intervals (95% CI) were computed using Poisson models. The trend of the number of screening invitations and mammograms and the trend of the number of cancer cases were reported by months. STATA/SE v. 16.1 was used for all analyses. In comparing data observed in 2020 with those obtained in 2019, we did not perform any formal statistical test of the hypotheses. In fact, we started from the fact that the pandemic had an impact on all human activity and particularly on the activity of health services; therefore, breast cancer diagnoses have been influenced by the pandemic. In these analyses, we describe how the timing of diagnoses was influenced and how this influenced the stage at diagnosis. We present *p*-values as a measure of the probability that a similar or larger difference would be observed by chance and 95% CI as a measure of uncertainty of the estimates. No significance threshold has been set.

Since we are comparing 2019 and 2020, it is relevant to consider how the date of diagnosis was collected. Usually, cancer registries follow the international cancer registration rules: the date of diagnosis coincides with the date of the first histological report or, failing this, with the date of the first hospitalization. The date of onset of symptoms or the date mammography is performed is never used because it is not provided for in the registration rules (https://www.encr.eu/sites/default/files/Recommendations/ENCR%20Recommendation%20DOI_Mar2022_0.pdf—accessed on 30 May 2022).

## 3. Results

A total of 1159 cancers are included in the study (589 in 2020 and 570 in 2019); there are 149 in situ tumours (76 cases in 2020 and 73 cases in 2019). The characteristics of the women included in the study are shown in Table 1. Most cancers are registered at ages 45–74 (screening range) with a modest increase in 2020 (70% vs. 63.6%). No changes were detected in morphology, pT, N, M, stage, and grading.

Stratifying by age (Table 2), in the 45–74-year-olds (screening range), there is a decrease in ductal tumours which is balanced by an increase in less defined forms, and a modest increase in T1 and no variation in terms of stage and grading.

Regarding the screening status, there is a significant decrease in interval cancers (26.9% vs. 22.6%) associated with an increase in ineligible patients (1.6% vs. 7.2%); 14 of 26 cancers in not eligible women in 2020 were diagnosed in women who missed screening due to suspension during lockdown. In young women <45, in 2020, there was a reduction in stage I (47.2% vs. 31.3%) and an increase in stage IV (3.8% vs. 10.4%), the differences were compatible with random fluctuations, while in 75+ women the stage distribution was mostly unchanged (slight increase in stage I and II and a slight decrease in stage III).

Concerning treatment (Table 3) in women with invasive breast cancer, in 2020, there was a significant reduction in surgical treatment (30.4 vs. 21.6%) associated with an increase in neoadjuvant treatment (19.1% vs. 22.8%). In this case, surgery can also be associated with hormone therapy, while neoadjuvant treatment only means chemotherapy.

Comparing the four periods of 2020 with respect to 2019, it can be observed that in the *pre-lockdown* period, there was a slight increase in new diagnoses in all age groups (Figure 1); during the *lockdown* a decrease in new diagnoses of cancer can be observed for all ages, which is greater for the age group 75+ [IRR 0.45 (95% CI 0.25–0.79)], and similar for the 45–74 age group, the screening target age and younger women, but for the latter the estimate is extremely imprecise (Figure 1). In the *low incidence* period, new diagnoses were recovered in 45–74 and in 75+ women while in the last period (*second wave*) there was a large increase in new diagnoses only for ages 45–74 [IRR 1.48 (95% CI 1.11–1.98)].

This trend in new cancer diagnoses fits well with the extension of the invitations to mammography screening and the number of mammograms performed (Figure 2). The number of new cancers (45–74-year-olds) closely follows the number of mammograms performed. In 2020, the suspension of invitations during the lockdown was followed by a modest reduction during the summer period (August) which, however, is shorter and less pronounced than the usual summer reduction in screening activities observed in 2019. Immediately after the summer, however, the resumption of invitations led to a rapid increase of cancer diagnoses in women aged 45–74.

## 4. Discussion

A decline in cancer diagnoses was observed in most countries where it has been investigated and for almost all tumour sites [10,11,14,24,25,26], in particular for those subject to screening [12,27,28,29,30,31]. The recovery of post-lockdown diagnosis almost never compensates for the previously observed decline [15,29,30,32].

Also, in our province in 2020, we recorded almost 700 fewer cases of cancer diagnoses compared to 2019: in particular, the decline concerned skin cancers, melanoma, bladder, prostate and colorectal but not breast cancers [15].

For the breast, our study confirms a decrease in cancer diagnoses during the lockdown period, followed by the identification of new cases in the following months, resulting, at the end of 2020, in a complete offset of the cases (+16 invasive tumours and +three tumours in situ).

Unlike what is reported in other countries, including Italian regions [18], we have not registered any overall impact on the early stages nor have we registered a shift towards advanced stages and we have seen a slight increase in metastatic forms.

By comparing the tumour size with those reported by AIOM (the Italian Association of Medical Oncology) in Italy [33], we observed a similar trend: a modest decrease in T1a and T1c tumours and a slight increase in T1b tumours, even if on the whole the registered tumours in our province represent smaller dimensions than the national and international data [33,34].

It is worth noting that the screening target age group, i.e., 45–74, showed an increase in diagnosed tumours (423 cases in 2020 and 377 cases in 2019: +43 invasive, +3 in situ) and no decrease in stage I at all. On the other hand, cancers in women out of the screening target age showed a decrease: in women aged <45, tumours declined, with a reduction in stage I (from 47.2% to 31.3%) and an increase in stage IV even if these differences in stage are compatible with random fluctuations. Even in women aged 75+, tumour incidence dropped in 2020: −22 invasive and +two in situ, without significant effects on the stage.

In general, these findings are consistent with a larger impact of the lockdown on opportunistic screening and case finding than on diagnoses made by the organized screening programme. In fact, thanks to the prompt resumption of screening and the larger volume of activities in summer and autumn, the screening target population has been invited for a mammography even if a few months later, and the number of screen detected cancers even increased in 2020. This last finding is not unexpected, since the average screening interval before mammograms in 2020 was slightly longer than in 2019, thus possibly favouring a higher detection rate.

In other studies, there was a decrease of in situ (−10% in Toss [18], −51% in Chou [31]) and early forms (−27% for stage I tumours in Chou [31]) and an increase in N+ and stage III [18,31], even if these were referring to a shorter period. The decrease in tumours in situ and in tumours with T1 was also observed by Vanni [19] and colleagues, also associated with an increase in N2 (14% in 2020 vs. 4% in 2019) and a slight increase in diameter (21 mm in 2020 compared to 16 mm in 2019).

In the Netherlands, a decline in in situ and stage I cancers was observed in 2020 (from 12% to 6% and from 40% to 36%, respectively) [20]. The suspension of screening programmes resulted in a decrease in screen-detected cancers [29,34], and an early tumour reduction, according to expectations but without a shift towards more advanced forms [35]. Even cancers diagnosed in out-of-range screening women do not appear to have been affected by the pandemic [29].

Regarding treatment, we observed a significant reduction in surgery in 2020 associated with an increase in neoadjuvants. A delay in treatment has also been observed in other countries [20,36] and in Italy [19] due to the increase of intensive care use and the consequent scarcity of anaesthetists to activate surgery rooms during the pandemic. Surprisingly, fewer studies observed an increase in neoadjuvant therapy [36], which was expected given the difficulties in performing timely surgery.

Among the limitations of this study, it is worth noting that our study is not able to detect small changes in the occurrence according to stage and the absence of information on biological variables (receptor state and Herb2). Among the strengths, we recall that it is a study on population data, so there are no sample selection biases. Furthermore, we only compared two years, thus increasing the influence of chance on our results. Nevertheless, the Table A1 shows that the magnitude of yearly fluctuations is usually small, and that 2019 was not a particularly low incidence year. Finally, data reporting and registration for 2020, particularly the last months, may be incomplete. This can explain some observed differences, such as the higher proportion of undetermined morphologies and the lower proportion of women receiving adjuvant chemotherapy in 2020.

## 5. Conclusions

In conclusion, in our study we did not observe any decrease in breast cancers diagnosed in 2020 compared to 2019. This result suggests that a prompt resumption of programmed screening may have limited the impact of the pandemic on the delay of breast cancer diagnoses at least in women aged 45–74. It would be interesting to see if some effects of the possible delay in diagnoses in women who are out of the screening target age or do not attend will become appreciable in the next few years.

## Figures and Tables

**Figure 1 cancers-14-03029-f001:**
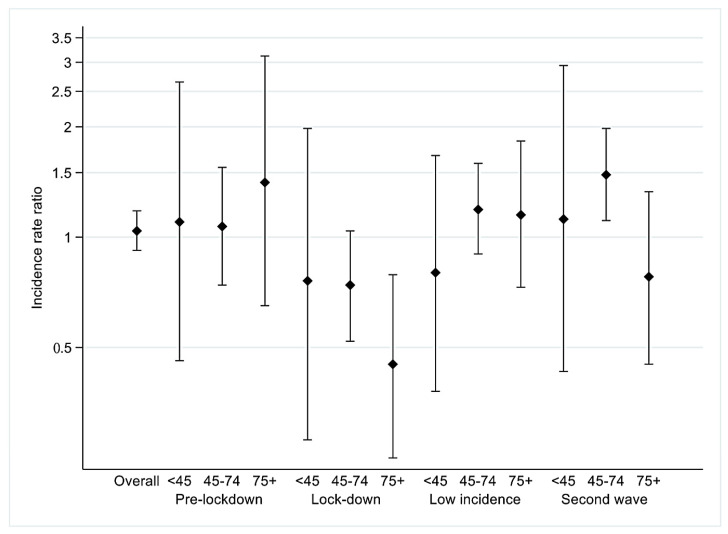
Incidence Rate Ratio of new cases in 2020 compared to 2019, by period and age group.

**Figure 2 cancers-14-03029-f002:**
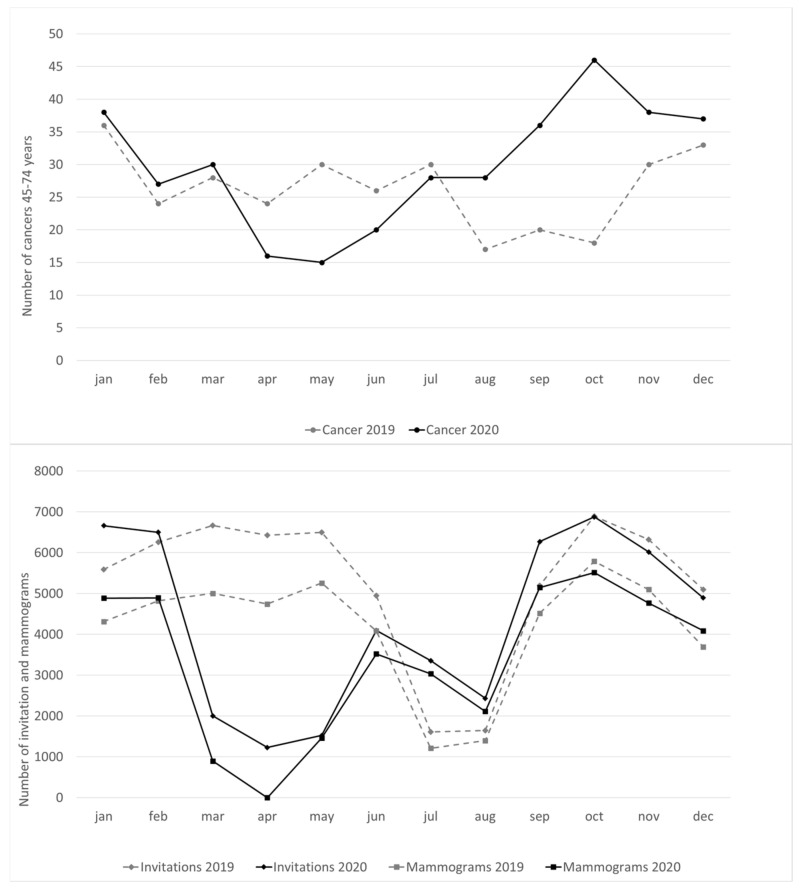
Number of cancers (**above**), and comparison with the number of invitations and mammograms (**below**), per month. Years 2019–2020.

**Table 1 cancers-14-03029-t001:** Characteristics of women with breast cancer, by year of diagnosis.

Variables	Year of Diagnosis					*p*-Value
	2019		2020		Total	
	n.	%	n.	%	n.	
**Overall**	**570**		**589**		**1159**	
Invasive cancers	497	87.2	513	87.1	1010	0.961
In situ	73	12.8	76	12.9	149	
**Only invasive cancers**						
**Age**						
<45	53	10.7	48	9.4	101	0.091
45–74	316	63.6	359	70.0	675	
75+	128	25.8	106	20.7	234	
**Morphology**						
ductal	395	79.5	412	80.3	807	0.212
lobular	77	15.5	65	12.7	142	
other	25	5.0	36	7.0	61	
**pT**						
T1	6	1.2	6	1.2	12	0.651
T1a	48	9.7	48	9.4	96	
T1b	91	18.3	106	20.7	197	
T1c	138	27.8	133	25.9	271	
T2	85	17.1	89	17.3	174	
T3	13	2.6	12	2.3	25	
T4	8	1.6	2	0.4	10	
Unknown	108	21.7	117	22.8	225	
**N**						
N0	347	69.8	365	71.2	712	0.879
N+	140	28.2	139	27.1	279	
Unknown	10	2.0	9	1.8	19	
**M**						
M0	484	97.4	494	96.3	978	0.31
M1	13	2.6	19	3.7	32	
**Stage**						
I	262	52.7	281	54.8	543	0.519
II	166	33.4	163	31.8	329	
III	52	10.5	43	8.4	95	
IV	13	2.6	19	3.7	32	
Unknown	4	0.8	7	1.4	11	
**Grading**						
1	30	6.0	26	5.1	56	0.917
2	322	64.8	335	65.3	657	
3	130	26.2	135	26.3	265	
Unknown	15	3.0	17	3.3	32	

**Table 2 cancers-14-03029-t002:** Characteristics of women with breast cancer, by year of diagnosis and age.

Variables	<45					45–74					75+				
	2019		2020			2019		2020			2019		2020		
	n	%	n	%	*p*-Value	n	%	n	%	*p*-Value	n	%	n	%	*p*-Value
**Overall**	**63**		**56**			**377**		**423**			**130**		**110**		
Invasive	53	84.1	48	85.7	0.809	316	83.8	359	84.9	0.683	128	98.5	106	96.4	0.300
In situ	10	15.9	8	14.3		61	16.2	64	15.1		2	1.5	4	3.6	
**Only invasive**															
**Morphology**															
ductal	45	84.9	44	91.7	0.535	259	82.0	285	79.4	0.046	91	71.1	83	78.3	0.107
lobular	3	5.7	2	4.2		51	16.1	54	15.0		23	18.0	9	8.5	
other	5	9.4	2	4.2		6	1.9	20	5.6		14	10.9	14	13.2	
**pT**															
T1	25	47.2	16	33.3	0.225	211	66.8	244	68.0	0.885	47	36.7	33	31.1	0.189
T2	8	15.1	10	20.8		47	14.9	51	14.2		30	23.4	28	26.4	
T3	2	3.8	0	0.0		7	2.2	9	2.5		4	3.1	3	2.8	
T4	0	0.0	0	0.0		0	0.0	1	0.3		8	6.3	1	0.9	
Unknown	18	34.0	22	45.8		51	16.1	54	15.0		39	30.5	41	38.7	
**N**															
N0	31	58.5	30	62.5	0.681	231	73.1	259	72.1	0.893	85	66.4	76	71.7	0.605
N+	22	41.5	18	37.5		81	25.6	94	26.2		37	28.9	27	25.5	
Unknown	0	0.0	0	0.0		4	1.3	6	1.7		6	4.7	3	2.8	
**M**															
M0	51	96.2	43	89.6	0.178	310	98.1	349	97.2		123	96.1	102	96.2	
M1	2	3.8	5	10.4		6	1.9	10	2.8		5	3.9	4	3.8	
**Stage**															
I	25	47.2	15	31.3	0.353	192	60.8	226	63.0	0.435	45	35.2	40	37.7	0.495
II	17	32.1	20	41.7		92	29.1	91	25.3		57	44.5	52	49.1	
III	8	15.1	6	12.5		25	7.9	27	7.5		19	14.8	10	9.4	
IV	2	3.8	5	10.4		6	1.9	10	2.8		5	3.9	4	3.8	
Unknown	1	1.9	2	4.2		1	0.3	5	1.4		2	1.6	0	0.0	
**Grading**															
1	4	7.5	1	2.1	0.175	22	7.0	22	6.1	0.280	4	3.1	3	2.8	0.305
2	28	52.8	19	39.6		211	66.8	236	65.7		83	64.8	80	75.5	
3	17	32.1	25	52.1		80	25.3	90	25.1		33	25.8	20	18.9	
Unknown	4	7.5	3	6.3		3	0.9	11	3.1		8	6.3	3	2.8	
**Screening**															
Screen-detected						199	63.0	228	63.5	0.003					
Interval cancers *						85	26.9	81	22.6						
non attenders						27	8.5	24	6.7						
not eligible						5	1.6	26	7.2						
**Total**	**53**		**48**			**316**		**359**			**128**		**106**		

* Interval cancers include all women with non-screen detected cancer and who had a negative screening test before cancer diagnosis, independently of the time elapsed since the negative test.

**Table 3 cancers-14-03029-t003:** Distribution of cases by treatments, surgery and chemotherapy and year of diagnosis.

Variables	2019		2020		Total	*p*-Value
	n.	%	n.	%	n.	
No treatment	40	8.0	48	9.4	88	0.016
Surgery only	151	30.4	111	21.6	262	
Surgery + adjuvant	211	42.5	237	46.2	448	
Neoadjuvant + surgery (done or not done yet)	95	19.1	117	22.8	212	
**Total**	**497**		**513**		**1010**	

## Data Availability

The data presented in this study are available on request from the corresponding author. The data are not publicly available due to ethical and privacy issues, requests of data should be approved by the Ethic Committee after the presentation of a study protocol.

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
