# Peer review of "Prompt Resumption of Screening Programme Reduced the Impact of COVID-19 on New Breast Cancer Diagnoses in Northern Italy"

_cancers, 2022, doi:10.3390/cancers14123029_

Round 1

Reviewer 1 Report

  • A brief summary

The paper provides an interesting snapshot on potential changes within breast cancer diagnosis in a region of high COVID incidence at the height of the first wave of the COVID pandemic. It aims to determine differences not only in incidence but more detailed pathologic features and the effect of suspension/reduction of the standard screening program. Its particular strength derives from a very detailed cancer registry which allows for high quality data with very few missing data points.

  • General concept comments

The concept is sound: comparing data year on year to look for significant differences but it is difficult to draw convincing conclusions that differences in data result from the effects of COVID, rather than being caused by annual fluctuation. This is particularly so when multiple comparisons are tested – I am not clear that the statistical methods are explicit in defining what methods have been used to account for multiple testing.

There is an inherent assumption that the incidence of invasive and in situ breast cancer is stable year on year (which may be correct) but it would be very helpful to see data from 3-5 additional earlier years [at least in terms of total numbers if not the detailed pathologic breakdown] so that trends and variances could be ascertained.

I was not clear on the method by which date of diagnosis was defined – typically in studies this is defined as the date of histological diagnosis. However, there are other potential points which could be used such as the date of detection or presentation with a symptomatic mass or the date of an abnormal screening mammogram. Relying only on the date of histologic diagnosis may lessen any true effect of COVID as there could be a ‘smoothing’ effect on the data by the suspension of diagnostic services beyond screening eg a positive screening mammogram way lead to biopsy at different interval according to the prevailing COVID conditions.

  • Specific comments 

L 72      are the (non-screening) data restricted to women only?

L 113    these are non-significant results with low patient numbers; it reads as if data are being used to support a predetermined theory of stage shift when they do not really support it

L108     Table 3 – the one significant difference is driven by a reduction in ductal subtype and corresponding rise in ‘other’ in the screened population. This seems to be a statistical fluke and may be driven by classification rather than any true phenomenon

L110     Table 3 – it was not exactly clear what “women with previous negative test” was defining – is this cases detected outside screening who had previously had negative screen? And non-attenders row is missing absolute values. Definition of non-attenders and not eligible is not clear.

L117     most of Table 2 already features in the excellent Table 1 and so is redundant. It could be replaced by a single line comment about T1a-c values (as at L107), though this difference itself is trivial and certainly of no clinical consequence.

L122     The increase in neoadjuvant treatment is surprising as chemotherapy rates surely declined – I presume this is therefore a term including neoadjuvant endocrine therapy and would be best to specify. It would be even more informative if the rates of neoadjuvant chemotherapy and endocrine therapy could be provided for each year. Given the high rates of surgery only (30-22%) it would seem that adjuvant endocrine treatment is not included as an adjuvant treatment so clarification for consistency would be helpful.

L144     Figure 2 may be more informative if the 2 years’ data are overlaid (or plotted as change in baseline numbers to avoid too complex a graph): in 2019 there is a dip in invitations and MMGs performed in July/Aug which is almost as dramatic as the dip in 2020 caused by COVID. It looks as though there is marked seasonal variation (ie summer holidays).

L146     The whole discussion section is quite long; a large proportion is spent re-presenting non-significant results from the results section and providing results from other studies. It would be more informative to perhaps focus more on why there were not significant differences and contrast that with other areas and postulate why that may be, particularly once addressing details about timing of diagnosis.

Author Response

Dear reviewer,

many thanks for the valuable comments. We have corrected all the things that you requested.

Best regards, 

Lucia Mangone

Reviewer 2 Report

The title of the article doesn't reflect their conclusion, because they conclude that pandemic had no or very little impact on the new breast cancer diagnoses. The title could improve.

They make a simple description of the statistical results obtained, without any reflexion on why it happened.

It could be appropiate adding more reflection about why the study didn't have an increase in number of advanced stage cancer, like it will appered in the other studies. It was only attribuited to the greast efficiency of their health system.

Also they could suggested that althought they didn't finde diferents in the diagnosis of new breast cancer during pandemic time, they would find change in the evolution of breast cancer in the next future years.

Although they comment that the lack of knowledge of instrisic subtpye is a limitation of the study, it colud be imporant reported this information. Currently in developing countries, the treatment of breast cancer is based on the inmunohistochemical results with or without molecular study as well.

Author Response

Dear reviewer,

many thanks for the valuable comments. We have corrected all things that you requested.

Best regards,

Lucia Mangone
